# *Klebsiella* and *Enterobacter* Isolated from Mangrove Wetland Soils in Thailand and Their Application in Biological Decolorization of Textile Reactive Dyes

**DOI:** 10.3390/ijerph17207531

**Published:** 2020-10-16

**Authors:** Aiya Chantarasiri

**Affiliations:** Faculty of Science, Energy and Environment, King Mongkut’s University of Technology North Bangkok, Rayong Campus, Rayong 21120, Thailand; aiya.c@sciee.kmutnb.ac.th; Tel.: +66-038-627-000 (ext. 5446)

**Keywords:** *Klebsiella pneumoniae*, *Enterobacter*, decolorization, ligninolytic activity, mangrove wetlands, reactive dye

## Abstract

Wastewater released from textile and dye-based industries is one of the major concerns for human and aquatic beings. Biological decolorization using ligninolytic bacteria has been considered as an effective and alternative approach for the treatment of dyeing wastewater. This study aimed to assess the isolation, characterization and application of soil bacteria isolated from mangrove wetlands in Thailand. Four active bacteria were genetically identified and designated as *Klebsiella pneumoniae* strain RY10302, *Enterobacter* sp. strain RY10402, *Enterobacter* sp. strain RY11902 and *Enterobacter* sp. strain RY11903. They were observed for ligninolytic activity and decolorization of nine reactive dyes under experimental conditions. All bacteria exhibited strong decolorization efficiency within 72 h of incubation at 0.01% (*w*/*v*) of reactive dyes. The decolorization percentage varied from 20% (C.I. Reactive Red 195 decolorized by *K. pneumoniae* strain RY10302) to 92% (C.I. Reactive Blue 194 decolorized by *Enterobacter* sp. strain RY11902) in the case of bacterial monoculture, whereas the decolorization percentage for a mixed culture of four bacteria varied from 58% (C.I. Reactive Blue 19) to 94% (C.I. Reactive Black 1). These findings confer the possibility of using these bacteria for the biological decolorization of dyeing wastewater.

## 1. Introduction

Reactive dyes are soluble chromophores comprising chemical reactive groups that chemically form covalent bonds with the hydroxyl groups of the cotton fiber under alkaline pH conditions [1,2]. Azo, anthraquinone and phthalocyanine dyes are the most widely used classes of reactive dyes in the color-based industries [3]. The dyes remain in the fiber after washing off, being considered dye-fixed on the fiber, and have fixation efficiencies in the range of 50–80% [2]. There are 20–50% of dye contents lost in the effluent depending on the class of the dye and the discharge to aquatic environments [2,4]. The wastewater released from fiber and textile manufacturing is incredibly complex in terms of volume and pollutant composition [5,6]. The industrialization of the dye-utilizing industries worldwide has been accompanied by a rise in wastewater pollution by dyestuffs, which have a significant effect on the surface water quality [7] and contaminate the surrounding environment as a consequence, such as ground water and soil. About 17–20% of freshwater pollution is caused by textile effluents that contain a large variety of dyes [8]. Azo dyes are problematic for biodegradability because of the electron-withdrawing groups in their structures [9,10]. Several studies have reported that azo dyes and their metabolites are toxic, carcinogenic and mutagenic [4,11]. The anthraquinone dyes are weakly toxic to aquatic organisms. However, they are highly resistant to biological degradation due to their fused aromatic structures [12]. Phthalocyanine dyes are metallic complexes and persist in biological degradation. The metal copper in their structure is potentially mutagenic and toxic [13]. The presence of these textile reactive dyes in aquatic ecosystems causes serious environmental problems and human health concerns.

The existing physicochemical and photochemical approaches for decolorizing dye effluents have the disadvantages of being expensive, being associated with operational difficulties and producing large amounts of polluting sludge [14,15]. Therefore, there is an urgent need for cost-effective, non-toxic and eco-friendly decolorizing methods. Biological decolorization by various microorganisms is considered as a potentially alternative approach [15,16]. Bacteria isolated from different sources can process the decolorization and complete mineralization of dyes under optimal conditions [16]. Several previous studies on the decolorization of textile reactive dyes have indicated the involvement of bacterial ligninolytic enzymes with oxidative activity, such as lignin peroxidase (LiP, EC 1.11.1.14), laccase (Lac, EC 1.10.3.2) and manganese peroxidase (MnP, EC 1.11.1.13) [14,17,18,19]. For this reason, the isolation of novel and effective decolorizing bacteria from various environments is an important challenge.

Mangrove ecosystems offer a promising habitat for the isolation of ligninolytic bacteria which are able to decolorize textile reactive dyes. Mangrove is a unique coastal wetland ecosystem that can be found along tropical and subtropical coastlines dominated by many halophilic plants [20,21]. They provide many ecological services such as the degradation of contaminants, nutrient cycling, nurseries for myriad aquatic fauna, pollution trapping, reduction of coastal erosion and the storage of surface runoff [21,22,23]. The mangrove soils are lignin-rich resources. Lignin is a complex heterogeneous biopolymer formed by phenylpropanoid units derived from three monolignols [24]. The source of lignin can be of plant origin that represents a significant part of plant litters input into the soil [25]. A previous report showed that there were four mangrove plants that had total lignin content more than that of the normal range of hardwood [26]. Moreover, mangrove soil is considered as a natural pollution trap for dyestuffs and related chemicals released from local communities, community enterprises and dye-utilizing industries. Previous studies found that several ligninolytic bacteria isolated from mangroves and aquatic ecosystems were indicated as dye-decoloring bacteria, such as *Lysinibacillus* sp. [14], *Paenibacillus* sp. [27] and *Roseobacter* sp. [28]. Most mangrove bacteria are naturally tolerant of saline conditions that are profitable for the treatment of dyeing wastewater. In dyeing processes, salt molecules help in the migration of dye molecules from the dye bath to the cotton fiber. They are usually released in the effluent and are harmful to freshwater ecosystems. However, little is known about the decolorization potential of mangrove bacteria.

This study aimed to isolate and characterize mangrove wetland bacteria from Rayong Province, Thailand, for the decolorization of textile reactive dyes by their ligninolytic enzymes. The selected bacteria were genetically identified using 16S rRNA nucleotide sequencing analysis. The active bacteria were observed for biological decolorization of the textile reactive dyes under experimental conditions. The knowledge gained from this study could possibly be used for wastewater treatment in dye-based industries.

## 2. Materials and Methods

### 2.1. Study Area

The study area was the “Phra Chedi Klang Nam” mangrove wetland in Rayong River Estuary, Rayong Province, Thailand (12°39′ N, 101°14′ E), which covers an area of 75,400 m^2^ that is dominantly grown over by mangroves *Avicennia* sp. and *Rhizophora* sp. [21,29]. Rayong Province is an important ecological area covered with mangrove wetlands and other aquatic ecosystems situated in the eastern region of the country. Several regions of this province are important agriculture, domestic and petrochemical industry zones for Thailand, which are situated near the riverine. Therefore, this estuarine has become chronically contaminated by many pollutants and dyestuffs, which have been trapped by the mangrove wetlands. The location of the study mangrove wetland is shown in Figure 1.

### 2.2. Sampling of Mangrove Wetland Soils

Soil samples from the mangrove wetland were collected during the late winter season in October 2016. Forty samples were taken randomly at a depth of 0–5 cm to obtain aerobic bacteria. Soil temperatures were measured at the site by a needle probe thermometer (Extech 39240, Extech Instruments, Waltham, MA, USA). The samples were kept at 4 °C and taken for isolation within 24 h of sampling.

### 2.3. Enrichment and Isolation of Bacteria from Mangrove Wetland Soils

The enrichment procedure for wetland bacteria and BSGYP culture medium for bacterial enrichment were previously described by Chantarasiri and Boontanom [14] and Chang et al. [30], respectively. The BSGYP medium contained 0.2 g of CaCl_2_·2H_2_O, 0.169 mg of CoCl_2_·6H_2_O, 0.085 mg of CuCl_2_·2H_2_O, 0.6 mg of H_3_BO_3_, 0.05 g of K_2_HPO_4_, 0.05 g of KH_2_PO_4_, 0.3 g of MgSO_4_·7H_2_O, 0.099 mg of MnCl_2_·4H_2_O, 0.1 g of NaCl, 0.22 mg of ZnCl_2_, 10.0 g of glucose, 0.18 g of yeast extract and 5.0 g of peptone in 1000 mL of distilled water. One gram of each collected mangrove soil sample was inoculated in 10 mL of BSGYP medium, pH 7.0. All cultures were incubated at 28 °C (the average temperature of sampling sites) for 48 h in an orbital shaking incubator at 120 rpm (LSI-3016R, Daihan Labtech, Gagok-ri, Korea). Subsequently, the culture media were serially diluted with sterilized 0.85% (*w*/*v*) NaCl solution supplemented with 0.1% (*w*/*v*) buffered peptone (Merck, Ponda, India) to obtain 1:10,000 dilutions. One hundred microliters of each diluted sample was spread plated on BSGYP medium agar and incubated at 28 °C for 24 h. The isolated bacteria were selected based on their morphological dissimilarities and named for the isolation numbers.

### 2.4. Screening of Dye-Decolorizing Bacteria by Rapid RBBR Method

The rapid screening method for the detection of ligninolytic bacteria was applied to screened dye-decolorizing bacteria. The rapid RBBR method described previously by Chang et al. [30] was used with slight modifications. Each isolated bacterium was inoculated and cultured in 5 mL of BSGYP medium supplemented with 0.01% (*w*/*v*) Remazol Brilliant Blue R reactive dye (RBBR, C.I. Reactive Blue 19, Sigma-Aldrich, Steinheim, Germany) at 28 °C for 48 h in the orbital shaking incubator at 120 rpm. All cultures were incubated in dark to avoid photo-decolorization of the dye. Bacterial cells were discarded from the culture medium by centrifugation at 10,000× *g* for 10 min (Digicen 21 R, Ortoalresa, Spain). The concentration of RBBR was determined in the cell-free supernatant using a spectrophotometer at 592 nm, which is the maximum wavelength (λ_max_) of RBBR. The uninoculated BSGYP medium supplemented with the respective dye was considered as a reference.

The decolorization percentage was calculated using the following formula: decolorization percentage (%) = ((Abs_i_ − Abs_f_)/Abs_i_) × 100. Abs_i_ is the initial absorbance of cell-free supernatant, while Abs_f_ is the final absorbance of cell-free supernatant after 48 h of incubation. All experiments were assayed in triplicate.

### 2.5. Growth Characterization of Dye-Decolorizing Bacteria

Dye-decolorizing bacteria were grown and characterized by culture in tryptone soya broth (HiMedia, Mumbai, India) under different pH, temperature and salinity conditions. The parameter values were: pH of 2.0–10.0, temperature of 25–50 °C and NaCl concentration of 0.05–5.00% (*w*/*v*). All cultures were incubated for 24 h in an orbital shaking incubator at 120 rpm. Bacterial growth was determined as optical density (OD) using a GENESYS 10S Vis spectrophotometer (Thermo Scientific, Waltham, MA, USA) at 600 nm after incubation. All experiments were assayed in triplicate.

### 2.6. Identification of Dye-Decolorizing Bacteria

Genetic identification of dye-decolorizing bacteria was performed by 16S rRNA gene sequence analysis. The genomic DNA of bacteria was extracted by the Genomic DNA Isolation Kit (Bio-Helix, Keelung City, Taiwan) according to the protocol described by Bio-Helix. Polymerase chain reaction (PCR) of 16S rRNA gene was carried out using the OnePCR reaction mixture (Bio-Helix, Keelung City, Taiwan) in the Mastercycler Nexus (Eppendorf, Hamburg, Germany). The universal primers used in PCR amplification were 27F (5′-AGAGTTTGATCMTGGCTCAG-3′) and 1492R (5′-TACGGYTACCTTGTTACGACTT-3′) (Sigma-Aldrich, Biopolis-Nucleos, Singapore) [31]. The PCR reaction was conducted according to Boontanom and Chantarasiri [32] and Chantarasiri [33]. PCR was performed for 35 amplification cycles, which involved a preheating step at 94 °C for 4 min, a denaturation step at 94 °C for 40 s, an annealing step at 55 °C for 60 s, an extension step at 72 °C for 1 min 10 s and a final extension step at 72 °C for 10 min. Approximately 1500-bp PCR products were analyzed using a 1.5% (*w*/*v*) OmniPur agarose gel (Calbiochem, Darmstadt, Germany) and visualized by Novel Juice (Bio-Helix, Keelung City, Taiwan). The PCR products were subsequently purified using the PCR Clean-Up Kit (Bio-Helix, Keelung City, Taiwan) and sequenced by the nucleotide sequencing service of Macrogen Inc. (Seoul, Korea). Sequence similarity analysis of the amplified 16S rRNA genes was aligned through the BLASTn program of the National Center for Biotechnology Information (NCBI) database [34]. The phylogenetic tree was inferred using the SeaView program version 5.0.2 [35] and visualized by the FigTree program version 1.4.4 (Institute of Evolutionary Biology, University of Edinburgh, Edinburgh, UK). The phylogenetic tree in terms of cladogram was generated by the BIONJ algorithm [36] with 100,000 bootstrap replications. All nucleotide sequences of 16S rRNA gene from this study were deposited in the GenBank database of NCBI under the accession numbers MT355793, MT355794, MT355797 and MT355798.

### 2.7. Preparation of Crude Peroxidases and Laccase

The crude ligninolytic enzymes with oxidative activity including peroxidases and laccase were prepared by culture of the effective bacterial isolates in BSGYP medium supplemented with 0.01% (*w*/*v*) Kraft lignin (Sigma-Aldrich, St. Louis, MO, USA). All dye-decolorizing bacteria were shake-flask cultured under the optimum growth conditions of each bacterium for 48 h in the orbital shaking incubator at 120 rpm. The bacterial cultures were centrifuged at 10,000× *g* at 4 °C for 10 min to obtain cell-free supernatants. The supernatants were concentrated by 10 kDa Amicon ultra centrifugal filter units (Millipore, Cork, Ireland) and the concentrated supernatants finally served as the crude enzyme solution. All crude enzymes were kept at 4 °C.

### 2.8. Determination of Bacterial Peroxidase and Laccase Activities

The activity of crude peroxidases and laccase was determined following the methods of Chang et al. [30] and Chantarasiri et al. [19]. Lignin peroxidase was measured by incubating 1.5 mL of crude enzyme solution with 0.5 mL of reaction mixture containing 2.5 mM 3,4-dimethoxybenzyl alcohol (Sigma-Aldrich, St. Louis, MO, USA) and 0.5 mM H_2_O_2_ (QRëC, New Zealand) in an assay buffer. The lignin peroxidase reaction was spectrophotometrically measured at 310 nm for 5 min. Manganese peroxidase activity was measured by incubating 1.5 mL of crude enzyme solution with 0.5 mL of reaction mixture containing 0.1 mM MnSO_4_, 0.1 mM H_2_O_2_ and 0.25 mM phenol red (LabChem, Zelienople, PA, USA) in an assay buffer. Manganese peroxidase reaction was spectrophotometrically measured at 610 nm for 5 min. Laccase activity was measured in 1 mL of reaction mixture by incubating 0.2 mL of crude enzyme solution with 75 mM 1,2-dihydroxybenzene (Sigma-Aldrich, St. Louis, MO, USA) in an assay buffer. Laccase reaction was spectrophotometrically measured at 440 nm for 10 min. The assay buffer used in this study was 50 mM sodium phosphate buffer at pH 7.0. One unit (U) of enzyme activity was defined as the amount of enzyme catalyzing in 1 µmol of substrates oxidized per minute of reaction. All experiments were assayed in triplicate.

### 2.9. Dye Decolorization Efficiency of Bacterial Monoculture and Mixed Culture

Dye decolorization efficiency was analyzed with nine textile reactive dyes as follows: C.I. Reactive Black 1, C.I. Reactive Brown 1, C.I. Reactive Blue 19, C.I. Reactive Blue 194, C.I. Reactive Blue 21, C.I. Reactive Green 19, C.I. Reactive Orange 122, C.I. Reactive Red 195 and C.I. Reactive Yellow 167 (INDAFIX, Bangkok, Thailand; Sigma-Aldrich, Steinheim, Germany). The maximum wavelength (λ_max_) of each reactive dye in this study was obtained by spectrophotometric analysis of a GENESYS 10S Vis spectrophotometer (Thermo Scientific, Waltham, MA, USA) in the wavelength scanning mode. The properties of the textile reactive dyes used in this study are shown in Table 1.

Each bacterial isolate was pre-cultured in tryptone soya broth under the optimum growth conditions for 24 h in the orbital shaking incubator at 120 rpm. Bacterial cells were harvested by centrifugation at 10,000× *g* at 4 °C for 10 min and then resuspended in sterilized 0.85% (*w*/*v*) NaCl to get the concentration of 0.5 McFarland standard (1.5 × 10^8^ cells/mL). These were used as the inoculums for bacterial monoculture. The inoculum of bacterial mixed culture was prepared by mixing all monoculture inoculum in a 1:1 ratio and resuspended to get the concentration of 0.5 McFarland standard.

A baffled flask containing 100 mL of BSGYP medium supplemented with 0.01% (*w*/*v*) textile reactive dye was seeded with 1% (*v*/*v*) of each bacterial inoculum. All monocultures were incubated under their optimum growth conditions for 72 h in the orbital shaking incubator at 120 rpm. The mixed culture was seeded and incubated under sharing growth conditions involving pH 8.0, 30 °C and 0.5% NaCl (*w*/*v*) for the aforementioned incubation time and agitation rate.

Five hundred microliters of culture suspension was aseptically collected to measure dye decolorization efficiency every 12 h. Decolorization of textile reactive dyes was observed in the cell-free supernatant using a spectrophotometer at λ_max_ of each dye. The decolorization percentage was calculated using the formula mentioned earlier. All experiments were performed in triplicate. The control was the BSGYP medium supplemented with 0.01% (*w*/*v*) textile reactive dye inoculated with *Escherichia coli* strain TISTR 073. This bacterium is a well-known enteric bacterium and is a member of the family Enterobacteriaceae in the class Gammaproteobacteria. It was employed in the decolorization of several commercial anthraquinone and azo dyes [37].

### 2.10. Data Analysis

The statistical analysis in this study was performed by one-way ANOVA followed by Tukey’s test with a 95% confidence interval (*p* < 0.05) using R software version 4.0.0 (R Foundation for Statistical Computing, Vienna, Austria).

## 3. Results and Discussion

### 3.1. Sampling of Mangrove Soils

Forty mangrove soil samples were randomly collected from the Phra Chedi Klang Nam mangrove wetland in Rayong Province, Thailand. The soil samples were likely rich in mangrove lignin due to the decayed mangrove plant litters. A previous report showed that several mangrove plants had higher total lignin content than that of other plants [26]. Moreover, the mangrove wetland soils have been impacted by chronic chemical and dyestuff contaminations from anthropogenic activities, as mentioned above in the “Introduction” section. The texture of all soil samples was muddy clay with a brownish-black color. In Thailand, most mangrove wetland soil is composed of fine clay particles and the color is grey-dark reddish brown [38]. The average temperature measured at the site using a needle probe thermometer was 28.10 ± 0.20 °C.

### 3.2. Enrichment and Isolation of Bacteria from Mangrove Soil

Soil bacteria were enriched, isolated and purified by the BSGYP medium. There were 190 dissimilar bacterial colonies isolated from the mangrove soil. Most isolated bacteria had white pigmentation, circular shape, entire margin and raised elevation. The morphology of isolated bacteria in terms of percentage is shown in Table 2.

### 3.3. Screening of Dye-Decolorizing Bacteria by Rapid RBBR Method

The bacterial isolates (*n* = 190) were screened for dye decolorization by the modified rapid RBBR method. They were cultured in the BSGYP medium amended with 0.01% (*w*/*v*) RBBR reactive dye (C.I. Reactive Blue 19). Almost all isolated bacteria showed vigorous growth in this BSGYP medium. However, they could not decolorize RBBR in the medium. The result revealed that only four bacterial isolates were able to decolorize RBBR with more than 20% of decolorization percentage. They were assigned as bacterium strains RY10302, RY10402, RY11902 and RY11903. The bacterial strain RY11903 exhibited maximum RBBR decolorization of 59.14 ± 0.87% within 48 h of incubation. All dye-decolorizing bacteria retained the original colony color, opaque white, after incubation in the BSGYP medium supplemented with RBBR. It was indicated that dye decolorization resulted from the enzymatic processes rather than cell adsorption of dye molecules [15].

### 3.4. Growth Characterization of Dye-Decolorizing Bacteria

The growth characteristics of four bacterial candidates were determined by their optical density values at 600 nm (OD_600_) after 24 h of incubation in tryptone soya broth under various conditions. The results showed that they could grow at pH values from 5.0 to 10.0, temperatures between 25 and 40 °C and NaCl concentrations of 0.5–5.0% (*w*/*v*). The optimum growth conditions of dye-decolorizing bacteria were different, as shown in Figure 2, Figure 3 and Figure 4. The optimal pH for bacterial growth ranged between weakly acidic and weakly alkaline. Therefore, the dye-decolorizing bacteria were considered to be neutrophilic bacteria. The optimal temperature for bacterial growth was 30 °C, except for the bacterial strain RY10302, which was 35 °C. All strains in this study were designated as mesophilic bacteria. The optimal pH and temperature for bacterial growth are reasoned by the climate conditions of mangrove wetlands. The pH of mangrove soil in Thailand is neutral or weakly acidic and the temperature ranges from 25.5 to 29.9 °C [38,39]. The optimal salinity for bacterial growth was 0.5–1.0% NaCl based on bacterial strains. The bacterial strain RY11903 optimally grew in 3.0% NaCl and could be considered as a slight halophilic bacterium.

The characteristics of textile industry wastewater vary according to the different processes, mills and countries. Yaseen and Scholz [40] reviewed the typical characteristics of real textile wastewater reported from many studies [41,42,43]. The dye effluents released from textile industries are mainly between a weakly acidic pH of 6.0 and an alkaline pH of 10.0. The temperature ranges from 35 to 40 °C. The salinity of textile wastewater after fiber scouring and rinsing processes is 2000 mg/L (0.2% *w*/*v*). Finally, the salinity increases after the soaping process to 5000 mg/L (0.5% *w*/*v*).

All growth characteristics of the four bacterial isolates were favorable in reported textile wastewater parameters. Therefore, these isolated dye-decolorizing bacteria were believed to be useful in the biological decolorization of real textile wastewater.

### 3.5. Genotypic Identification of Dye-Decolorizing Bacteria

Dye-decolorizing bacteria were genetically identified by PCR amplification and nucleotide analysis of 16S rRNA gene. Alignment using the BLASTn program revealed that the bacterial strain RY10302 was closely similar to *Klebsiella pneumoniae* strain DSM 30104 with 97% query coverage and 99.02% identity. Three strains of dye-decolorizing bacteria, including strains RY10402, RY11902 and RY11903, were closely similar to *Enterobacter tabaci* strain YIM Hb-3 with 98–100% query coverage and 97.38–98.36% identity. The genotypic identification of dye-decolorizing bacteria is shown in Table 3. The E values of all BLASTn alignment results were zero. A phylogenetic tree in terms of cladogram using the BIONJ algorithm with 100,000 bootstrap replications is shown in Figure 5. All nucleotide sequences of 16S rRNA genes from this study were deposited in the GenBank database of NCBI under the accession numbers MT355793, MT355794, MT355797 and MT355798, as previously mentioned in the “Materials and Methods” section. Indeed, the bacterial strain RY10302 was confidently designated as *K. pneumoniae* strain RY10302. The other three strains were designated in the genus level as *Enterobacter* sp. due to their percentages of identity obtained from BLASTn program being lower than 99%. These dye-decolorizing bacteria were categorized in class Gammaproteobacteria of phylum Proteobacteria. All bacteria were stored as frozen stocks in 15% (*v*/*v*) glycerol and kept at the Faculty of Science, Energy and Environment, King Mongkut’s University of Technology North Bangkok, Thailand.

*Klebsiella* and *Enterobacter* are considered as members of total coliforms belonging to the family Enterobacteriaceae in the class Gammaproteobacteria. *K. pneumoniae* is an opportunistic pathogen causing nosocomial infections. It is widespread in various environments such as drinking water, industrial effluent, plants, sewage, soil and surface water [44]. It could be isolated from water and sediment samples from tropical estuaries and mangrove wetland ecosystems [44,45]. It complies with the habitat of *K. pneumoniae* strain RY10302 isolated from mangrove wetland soil in this study. *Enterobacter* species are common nosocomial pathogenic bacteria [46]. They are distributed in a diverse range of environments including soil, vegetation, water and human feces [47]. The most closely related *Enterobacter* of the three isolated bacteria in this study was *E. tabaci* strain YIM Hb-3. It was first isolated from the stems of cultivated tobacco (*Nicotiana tabacum* L.) collected from China in 2015 [48]. The 16S rRNA gene sequence of *E. tabaci* strain YIM Hb-3 was closely similar to six species of *Enterobacter* including *Enterobacter mori*, *Enterobacter asburiae*, *Enterobacter hormaechei*, *Enterobacter ludwigii*, *Enterobacter cloacae* and *Enterobacter cancerogenus*. Several *Enterobacter* species could be isolated from the rhizosphere of black mangroves (*Avicennia germinans* L.) [49] and the branches of red mangroves (*Rhizophora mangle*) [50]. They were considered as phosphate-solubilizing bacteria and mangrove-associated endophytic bacteria. It likely conforms to the nature and habitat of isolated *Enterobacter* sp. strains RY10402, RY11902 and RY11903.

### 3.6. Determination of Bacterial Peroxidase and Laccase Activities

Ligninolytic enzymes are extracellular oxidative enzymes commonly produced from white-rot fungi and primarily include lignin peroxidases, manganese peroxidases, versatile peroxidases (VP, EC 1.11.1.16) and laccases. Only very recently, dye-decolorizing peroxidases (DyPs, EC 1.11.1.19) have been identified in some ligninolytic fungi and bacteria [25]. However, knowledge about this group of peroxidases is very limited [51]. These ligninolytic enzymes synergistically and efficiently degrade non-phenolic and phenolic subunits of lignin polymer structure [52,53]. The ligninolytic enzymes can oxidize various pollutants such as aromatic dyes, chlorophenols, polycyclic aromatic hydrocarbons (PAHs), synthetic dyes and related model compounds [54,55,56]. Ligninolytic bacteria and their enzymes have been employed as a potentially alternative approach and cost-effective method for the bioremediation of textile dyes. Unfortunately, bacterial ligninolytic enzymes have not been studied extensively and the ligninolytic species are limited. They have only been found among some proteobacteria and actinomycetes [18].

In this study, the four dye-decolorizing bacteria of Gammaproteobacteria were examined for peroxidase and laccase activities. The results showed that *K. pneumoniae* strain RY10302 was not a peroxidase-producing bacterium. It could only produce laccase with enzymatic activity of 0.45 U/mL. The enzymatic assays exhibited that the three strains of *Enterobacter* sp. could yield 0.13–0.86 U/mL of lignin peroxidase activity and 0.66–1.47 U/mL of laccase activity. *Enterobacter* sp. strain RY11903 was considered as the most effective bacterium with significant lignin peroxidase and laccase activities among the four dye-decolorizing bacteria. All dye-decolorizing bacteria isolated from the mangrove wetland ecosystem were designated as laccase-producing bacteria. Interestingly, they could not produce manganese peroxidase. Their peroxidase and laccase activities are shown in Table 4.

Immobilized ligninolytic enzymes offer greater decolorization of dye-based pollutants in textile effluents and are considered as an environmental responsive technology [57]. This innovation requires purified ligninolytic enzymes. Therefore, the purification and polishing of ligninolytic enzymes are suggested for further study.

Various mangrove-isolated bacteria are able to produce ligninolytic enzymes because they are adapted to the lignin-rich soil found in mangroves and related aquatic ecosystems. An actinomycete bacteria, *Streptomyces psammoticus* strain NJP 49, was isolated from mangrove sediments in the west coast of India. It was capable of producing the three major ligninolytic enzymes, including approximately 2.00 U/mL of lignin peroxidase, 0.25 U/mL of manganese peroxidase and 1.50 U/mL of laccase [55]. A recent study found that *Lysinibacillus sphaericus* strain JD1103, isolated from a soil sample in the Rayong River Estuary in Thailand, near the sampling area in this study, had ligninolytic activity of crude lignin peroxidase by 0.90 U/mL and crude laccase by 1.13 U/mL [14].

Knowledge of *K. pneumoniae* and *Enterobacter* sp. isolated from mangrove soil with their ligninolytic activity remains limited. Besides the mangrove environment, there are several studies reported for ligninolytic *K. pneumoniae*. *K. pneumoniae* strain IITRC13 was isolated from the pulp industry in India and found to be associated with lignin degradation by its lignin peroxidase, manganese peroxidase and laccase [58]. *K. pneumoniae* strain 601 isolated from soil in China showed laccase activity with both thermal and broad pH stability [59]. *K. pneumoniae* strain NX-1 isolated from leaf mold samples in China could efficiently produce lignin peroxidase and laccase [60]. Four strains of *K. pneumoniae* isolated from the pulp and paper industry in India were designated as manganese peroxidase and laccase producing bacterium [61]. It is evidenced that *Klebsiella* species represent a promising source of ligninolytic enzymes. However, lignin and manganese peroxidase activities were not detected in *K. pneumoniae* strain RY10302, as shown in Table 4.

Ligninolytic *Enterobacter* species were isolated from various soil samples. *Enterobacter lignolyticus* strain SCF1 isolated from tropical forest soil in the USA was determined to be a ligninolytic bacterium. Its genome revealed two putative laccases and a putative peroxidase [62]. A laccase-producing bacterium was isolated from soil and decaying wood samples in Nigeria and molecularly identified as *E. ludwigii* [63]. *E. cloaceae* strain IITRCS11 was isolated from a dumping site in India. The function of ligninolytic enzymes was observed in decolorization and showed a high content of manganese peroxidase and laccase [64]. *E. cancerogenus* and *E. ludwigii* were isolated from petroleum-contaminated soil and animal manure samples in Turkey. They showed ligninolytic activities by degradation of Kraft lignin and RBBR decolorization [65]. Several *Enterobacter* species were employed in ligninolytic applications and showed successful performance. In this study, three isolated *Enterobacter* strains did not produce manganese peroxidase, as shown in Table 4.

The lack of peroxidases in the environmental strains from the studied mangrove wetlands is not well-understood. It might be explained by the inappropriate conditions for peroxidase preparations of *Klebsiella* and *Enterobacter* strains in this study. According to a recent previous study, it was reported that ligninolytic *Streptomyces* sp. strain S6 produced the highest enzyme activity for lignin peroxidase, slight laccase activity and no manganese peroxidase activity when grown on W minimal medium supplemented with Kraft lignin [66]. Likewise, several strong lignin degraders only produced manganese peroxidase and laccase, and no lignin peroxidase activity was detected when bacteria such as *Pandoraea* strain sp. B-6, *Comamonas* sp. strain B-9 and *Novosphingobium* sp. strain B-7 were grown on culture supplemented with Kraft lignin [67,68,69].

### 3.7. Dye Decolorization Efficiency of Bacterial Monoculture and Mixed Culture

The dye decolorization efficiency of the four dye-decolorizing bacteria were studied in monoculture and mixed-culture systems under experimental conditions. The final reactive dye concentration used in this study was 0.01% (*w*/*v*) amended in BSGYP medium. It was caused by a dye mass concentration suspended in the textile effluent that reported up to 50 mg/L (0.005%, *w*/*v*) [15]. Satisfactory results of dye decolorization by monoculture and mixed-culture bacteria were obtained after 36 h of incubation (Figure 6A–E).

In the monoculture experiment, all dye-decolorizing bacteria were able to decolorize nine reactive dyes by more than 20% within 72 h (Figure 6A–D), while the control bacterium, *E. coli* strain TISTR 073, showed a slight level of decolorization (Figure 6F). Three strains of *Enterobacter* evidentially had better dye decolorization efficiency than *K. pneumoniae* strain RY10302 at the same incubation time. *Enterobacter* sp. strain RY11902 was considered as the most effective decolorizing bacterium in this study because it productively decolorized four reactive dyes including C.I. Reactive Black 1 (azo), C.I. Reactive Blue 194 (azo), C.I. Reactive Blue 21 (phthalocyanine) and C.I. Reactive Green 19 (azo) by 88–92% within 72 h (Figure 6C and Table 5). *Enterobacter* sp. strain RY11903 was the second most effective decolorizing bacterium. It could decolorize C.I. Reactive Black 1, C.I. Reactive Blue 194, C.I. Reactive Blue 21 and C.I. Reactive Green 19 by 70–81% within 72 h (Figure 6D and Table 5). The decolorization efficiency of *Enterobacter* strains 11902 and 11903 conformed to their lignin peroxidase and laccase activities, as mentioned above in Table 4.

Interestingly, the mixed culture of dye-decolorizing bacteria was found to decolorize more efficiently than monoculture. It could decolorize six reactive dyes at 83–94% within 72 h, including C.I. Reactive Black 1, C.I. Reactive Brown 1 (azo), C.I. Reactive Blue 194, C.I. Reactive Blue 21, C.I. Reactive Green 19 and C.I. Reactive Red 195 (azo) (Figure 6E, Table 5). According to Karim et al. [15], the decolorization percentage of textile reactive dyes by bacterial monoculture and mixed culture was reported. The results showed 0–90% of dye decolorization in the case of bacterial monoculture and 65–90% of dye decolorization in the case of mixed culture. It was believed that bacterial species in the mixed culture synergistically interacted and co-metabolized to decolorize the reactive dye [70,71]. The different bacterial strains have different ability to break down the dye structure. Alternatively, they may use some harmful intermediates produced by co-existing strains for further degradation [72,73]. Therefore, the mixed culture of dye-decolorizing bacteria should be employed in the biological decolorization of various reactive dyes in textile wastewater. However, the compatibility of bacterial strains in the mixed culture should be of concern and antagonistic bacteria need to be avoided.

C.I. Reactive Black 1 was the most decolorized dye. It was decolorized by the mixed culture of dye-decolorizing bacteria by 94.20 ± 0.83% within 72 h (*p* < 0.0001) (Table 5). The second and third most decolorized dyes were C.I. Reactive Blue 194 and C.I. Reactive Green 19, respectively. They were decolorized by *Enterobacter* sp. strain RY11902 by 92.45 ± 0.40% (*p* < 0.001) and the mixed bacterial culture by 92.05 ± 1.78% (*p* < 0.0001) within 72 h (Table 5). Finally, the study found that C.I. Reactive Blue 19 (anthraquinone), C.I. Reactive Orange 122 (azo) and C.I. Reactive Yellow 167 (azo) were persistent in the decolorization of both bacterial monoculture and mixed culture. The differences in the decolorization pattern for each reactive dye is due to the dissimilarity in specificities, structure and complexity of dyes [15].

Textile reactive dyes in this study were believed to be decolorized by bacterial lignin peroxidase and laccase. Lignin peroxidase is an important enzyme for the treatment of colored industrial effluents and other xenobiotics [74]. It has been used to oxidize a variety of recalcitrant aromatic compounds including dyes in the class of azo [75]. Lignin peroxidase absolutely needs H_2_O_2_ as a terminal electron acceptor in decolorizing reactions [7]. Laccase could oxidize lignin, aromatic amines, polyphenols and related aromatic compounds. It has broad substrate specificity and is capable of decolorizing wastewater containing structurally different azo dyes [76]. Laccase requires oxygen and redox mediators during enzyme mechanisms [7]. It was demonstrated by relatively complete decolorization of many azo-reactive dyes in this study, such as C.I. Reactive Black 1, C.I. Reactive Blue 21 and C.I. Reactive Green 19. The strength of these enzymes is in the non-generation of hazardous intermediates and toxic byproducts after decolorizing, such as toxic aromatic amines [7]. Therefore, the byproducts and metabolites formed as a result of dye decolorization by bacteria were found to be less toxic compared with untreated wastewater.

Many reactive dyes were studied for biological decolorization by various effective dye-decolorizing bacteria, as shown in Table 6. The mixed cultures of four bacterial strains in this study are promising candidates for the decolorization of C.I. Reactive Black 1 and C.I. Reactive Green 19 when compared to previous studies. They have also been used successfully for the decolorization of C.I. Reactive Red 195 in weakly alkaline wastewater supplemented with a high concentration of the reactive dye. *Enterobacter* sp. strain RY11903 is suggested for the decolorization of C.I. Reactive Blue 19 in saline-alkaline wastewater since other bacteria in several studies favored decolorizing under weakly acidic wastewater without salt supplementation.

Finally, *K. pneumoniae* strain RY10302, *Enterobacter* sp. strain RY10402, *Enterobacter* sp. strain RY11902 and *Enterobacter* sp. strain RY11903 should be employed for wastewater treatment in dye-based industries. The growth characteristics of these bacteria are suitable for real textile wastewater. A proposed wastewater-treatment process uses an aerobic basin and related aerobic methods due to the limitation of their oxidative ligninolytic enzymes. Carbon and nitrogen sources are indicated as the obligate requirement for bacterial growth and the induction of dye decolorization [15]. A previous study reported that only a few bacteria are capable of utilizing dyes as a sole carbon source [82]. Nutrient feeding units are recommended for the proposed wastewater-treatment process. A major concern for bacterial application is clinical awareness because *K. pneumoniae* and *Enterobacter* sp. are considered as pathogens capable of causing nosocomial infections. Therefore, the effluent should be disinfected using chlorine or ultraviolet light after decolorization.

## 4. Conclusions

A variety of reactive dyes released by the textile and dye-based industries are of significant concern for human beings and living organisms in aquatic environments. The mangrove wetland is a unique ecosystem rich in mangrove lignin. It is recorded as a potential source for the isolation of ligninolytic bacteria. Ligninolytic bacteria are able to decolorize various textile reactive dyes by the oxidative reactions of peroxidases and laccase. Therefore, ligninolytic bacteria are promising candidates for the biological decolorization of reactive dyes. In this study, there were four strains of effective ligninolytic bacteria isolated from the Phra Chedi Klang Nam mangrove wetland in Thailand. Their growth characteristics possibly thrived under textile wastewater parameters. They were genetically identified by 16S rRNA nucleotide sequencing and assigned as *K. pneumoniae* strain RY10302, *Enterobacter* sp. strain RY10402, *Enterobacter* sp. strain RY11902 and *Enterobacter* sp. strain RY11903. Interestingly, they could only produce lignin peroxidase and laccase. These ligninolytic enzymes were believed to be responsible for the decolorization of reactive dyes. All bacterial strains exhibited decolorization of nine reactive dyes within 72 h, both in the case of bacterial monoculture and mixed culture experiments. Finally, the four dye-decolorizing bacteria obtained in this study could possibly be used in the biological decolorization of textile reactive dyes. Further studies are suggested concerning the purification of ligninolytic enzymes, enzyme immobilization for wastewater treatment applications, identification and toxicity of metabolizes after decolorization reactions and decolorization experiments under a natural environment.

## Figures and Tables

**Figure 1 ijerph-17-07531-f001:**
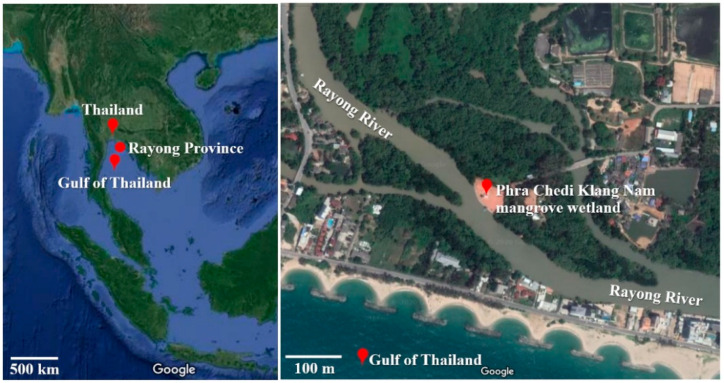
Location of the Phra Chedi Klang Nam mangrove wetland situated in Rayong Province, Thailand. (12°39′ N, 101°14′ E) (Source: Google Maps).

**Figure 2 ijerph-17-07531-f002:**
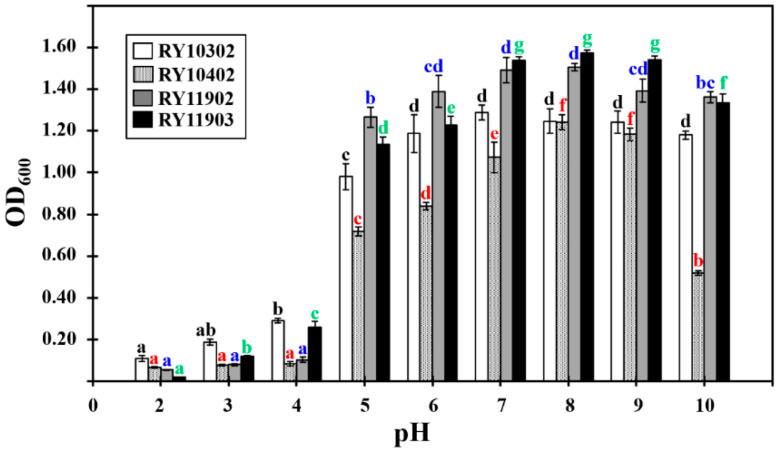
Effect of pH on the growth of four dye-decolorizing bacteria expressed as optical density measured at 600 nm. All experiments were performed in triplicate. Different letters with the same color on the top of SD bars indicate significant differences (*p* < 0.05).

**Figure 3 ijerph-17-07531-f003:**
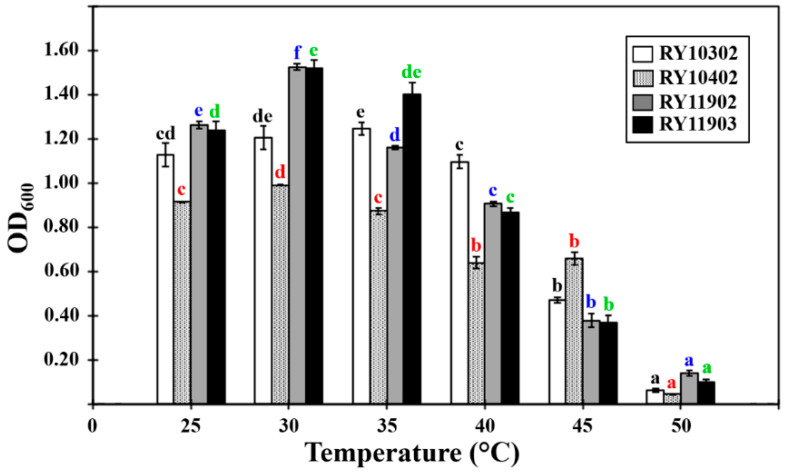
Effect of temperature on the growth of four dye-decolorizing bacteria expressed as optical density measured at 600 nm. All experiments were performed in triplicate. Different letters with the same color on the top of SD bars indicate significant differences (*p* < 0.05).

**Figure 4 ijerph-17-07531-f004:**
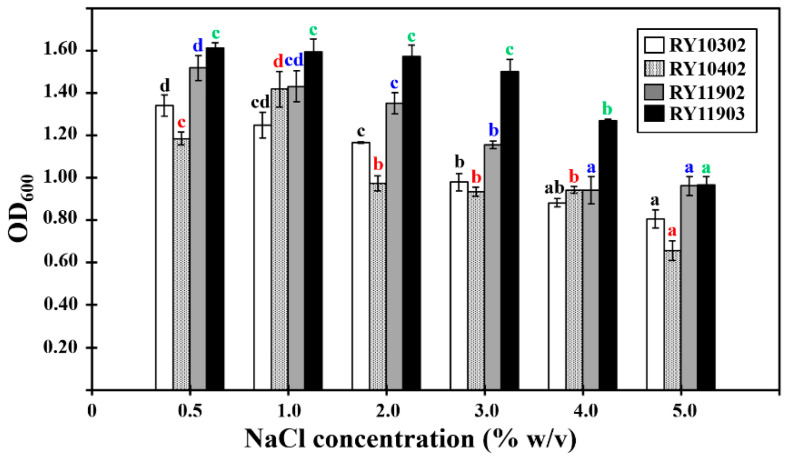
Effect of salinity on the growth of four dye-decolorizing bacteria expressed as optical density measure at 600 nm. All experiments were performed in triplicate. Different letters with the same color on the top of SD bars indicate significant differences (*p* < 0.05).

**Figure 5 ijerph-17-07531-f005:**
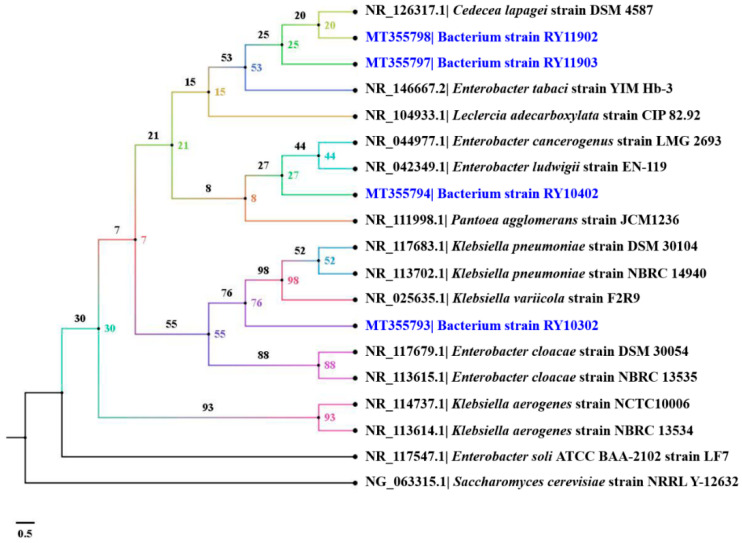
Phylogenetic tree is shown in terms of cladogram of four dye-decolorizing bacteria using the BIONJ algorithm with 100,000 bootstrap replications. The phylogenetic tree was generated by the SeaView program version 5.0.2 and visualized by the FigTree program version 1.4.4.

**Figure 6 ijerph-17-07531-f006:**
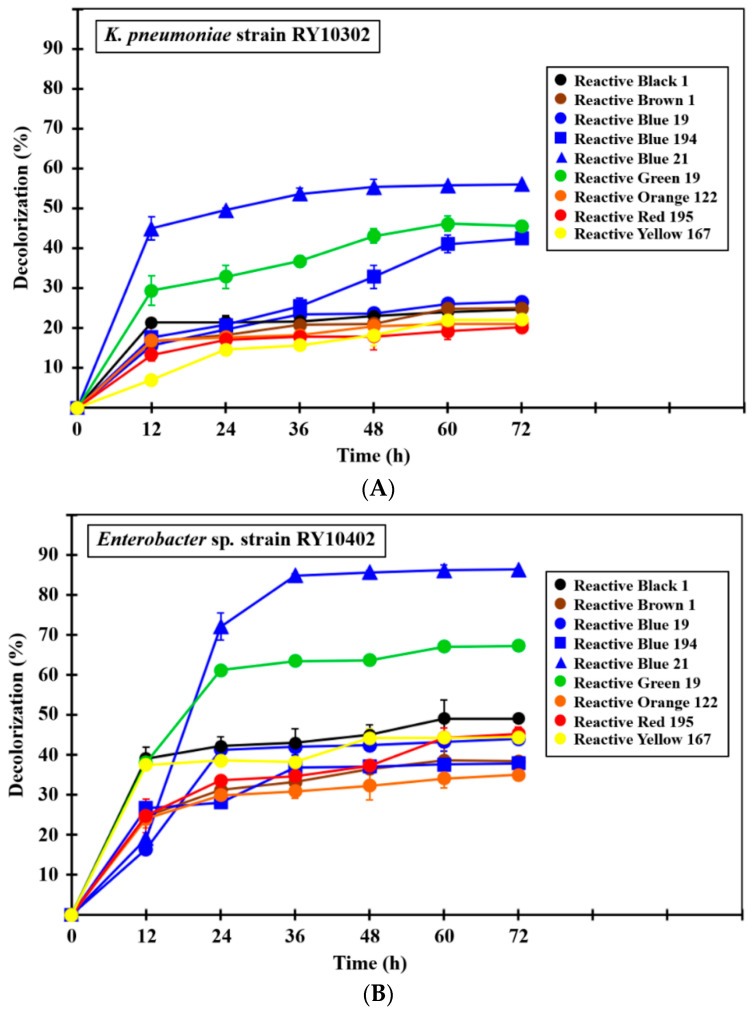
Decolorization of nine textile reactive dyes by various dye-decolorizing bacteria under experimental conditions. (**A**) Dye decolorization by *Klebsiella pneumoniae* strain RY10302. (**B**) Dye decolorization by *Enterobacter* sp. strain RY10402. (**C**) Dye decolorization by *Enterobacter* sp. strain RY11902. (**D**) Dye decolorization by *Enterobacter* sp. strain RY11903. (**E**) Dye decolorization by mixed bacterial culture of four isolated strains. (**F**) Dye decolorization by *Escherichia coli* strain TISTR 073 (a control dye-decolorizing bacterium). All experiments were performed in triplicate.

**Table 1 ijerph-17-07531-t001:** Properties of the nine reactive dyes used in this study.

Textile Reactive Dye	Class of Textile Reactive Dye	λ_max_ (nm)
C.I. Reactive Black 1	Azo	606
C.I. Reactive Brown 1	Azo	489
C.I. Reactive Blue 19	Anthraquinone	592
C.I. Reactive Blue 194	Azo	594
C.I. Reactive Blue 21	Phthalocyanine	670
C.I. Reactive Green 19	Azo	636
C.I. Reactive Orange 122	Azo	489
C.I. Reactive Red 195	Azo	544
C.I. Reactive Yellow 167	Azo	408

**Table 2 ijerph-17-07531-t002:** Percentage of the morphology of the isolated bacteria from the mangrove wetland soil.

Pigmentation (Percentage)		Shape (Percentage)		Margin (Percentage)		Elevation (Percentage)	
White	49.47	Circular	94.22	Entire	71.57	Raised	42.63
Yellow	23.16	Filamentous	3.68	Erose	12.11	Convex	42.11
Red	10.00	Irregular	1.58	Undulate	11.58	Umbonate	10.52
Colorless	17.37	Punctiform	0.52	Filamentous	4.74	Flat	4.74
Total	100.00	Total	100.00	Total	100.00	Total	100.00

**Table 3 ijerph-17-07531-t003:** Identity percentage of 16S rRNA gene sequences for the four dye-decolorizing bacteria.

Bacterial Isolate	Close Bacteria	GenBank Accession Number (References)	Query Cover (%)	Identity (%) *	GenBank Accession Number (Deposited)
**RY10302**	*Klebsiella pneumoniae*	NR_117683.1	97	99.02	MT355793
	strain DSM 30104				
**RY10402**	*Enterobacter tabaci*	NR_146667.2	98	97.38	MT355794
	strain YIM Hb-3				
**RY11902**	*Enterobacter tabaci*	NR_146667.2	99	97.91	MT355798
	strain YIM Hb-3				
**RY11903**	*Enterobacter tabaci*	NR_146667.2	100	98.36	MT355797
	strain YIM Hb-3				

Note: * Identity results were analyzed on 19 April 2020.

**Table 4 ijerph-17-07531-t004:** Peroxidase and laccase activities of the four dye-decolorizing bacteria.

Bacterial Strain	Enzymatic Activities (U/mL)		
	Lignin Peroxidase	Manganese Peroxidase	Laccase
***K. pneumoniae***	No activity	No activity	0.45 ± 0.03 ^a^ (*p* < 0.001)
**strain RY10302**			
***Enterobacter* sp.**	0.13 ± 0.02 ^a^ (*p* < 0.001)	No activity	0.81 ± 0.02 ^c^ (*p* < 0.001)
**strain RY10402**			
***Enterobacter* sp.**	0.62 ± 0.03 ^b^ (*p* < 0.001)	No activity	0.66 ± 0.02 ^b^ (*p* < 0.001)
**strain RY11902**			
***Enterobacter* sp.**	0.86 ± 0.06 ^c^ (*p* < 0.001)	No activity	1.47 ± 0.03 ^d^ (*p* < 0.001)
**strain RY11903**			

Note: The mean values in the same enzymatic activity followed by the same letter were not significantly different according to Tukey’s test (*p* < 0.05) among the bacteria. All experiments were assayed in triplicate.

**Table 5 ijerph-17-07531-t005:** Decolorization of textile reactive dyes by various dye-decolorizing bacteria under experimental conditions at 72 h of incubation.

Bacterial Strain	Decolorization of Textile Reactive Dye (%) at 72 h of Incubation								
	Black 1	Brown 1	Blue 19	Blue 194	Blue 21	Green 19	Orange 122	Red 195	Yellow 167
***K. pneumoniae***	24.65 ± 0.74 ^b^	24.96 ± 0.26 ^b^	26.51 ± 0.44 ^b^	42.43 ± 0.35 ^b^	55.97 ± 0.05 ^b^	45.49 ± 0.60 ^b^	21.00 ± 0.01 ^b^	20.10 ± 0.00 ^b^	22.00 ± 0.01 ^b^
**strain**									
**RY10302**									
***Enterobacter* sp.**	49.00 ± 0.00 ^c^	38.30 ± 0.26 ^c^	43.94 ± 0.07 ^d^	37.83 ± 0.29 ^b^	86.23 ± 0.32 ^d^	67.23 ± 0.40 ^c^	34.96 ± 1.12 ^c^	45.23 ± 1.71 ^c^	44.29 ± 0.66 ^c^
**strain**									
**RY10402**									
***Enterobacter* sp.**	89.52 ± 1.41 ^e^	66.27 ± 4.70 ^d^	37.27 ± 3.96 ^c^	92.45 ± 0.40 ^d^	88.67 ± 0.42 ^e^	91.09 ± 0.68 ^e^	54.50 ± 2.17 ^e^	60.27 ± 2.18^d^	40.86 ± 3.71 ^c^
**strain**									
**RY11902**									
***Enterobacter* sp. strain**	73.85 ± 3.18 ^d^	72.32 ± 1.71 ^d^	60.34 ± 0.94 ^e^	69.91 ± 5.24 ^c^	82.70 ± 0.76 ^c^	81.35 ± 0.84 ^d^	40.38 ± 2.02 ^d^	64.22 ± 2.48 ^d^	50.17 ± 2.12 ^d^
**RY11903**									
**Mixed**	94.20 ± 0.83 ^f^	83.03 ± 2.36 ^e^	58.25 ± 1.86 ^e^	86.81 ± 1.19 ^d^	88.32 ± 0.67 ^de^	92.05 ± 1.78 ^e^	63.01 ± 1.10 ^f^	84.32 ± 1.03 ^e^	68.26 ± 1.37 ^e^
**culture**									
***E. coli***	4.67 ± 0.22 ^a^	12.38 ± 0.44^a^	16.56 ± 2.91 ^a^	5.92 ± 0.81 ^a^	13.37 ± 1.70 ^a^	14.53 ± 0.36 ^a^	15.10 ± 0.94 ^a^	11.07 ± 0.81 ^a^	14.36 ± 1.07 ^a^
**strain**									
**TISTR 073**									

Note: The mean values in the same column (decolorization efficiency of reactive dye) followed by the same letter were not significantly different according to Tukey’s test (*p* < 0.05) among the bacteria. All experiments were assayed in triplicate.

**Table 6 ijerph-17-07531-t006:** Decolorization of some reactive dyes by isolated decolorizing-bacteria and other bacteria isolated from various environments.

Reactive Dye	Decolorizing Bacteria	Source of Isolation	Dye Concentration /Decolorization Percentage	Conditions for Decolorization	References
**C.I. Reactive Black 1**	Mixed bacterial culture	Dye contaminated soils, India	100 mg/L,	37 °C, 12 h	[73]
	(*Lysinibacillus* sp. strains BAB-4931 and BAB-4935, *Raoultella* sp. strain BAB-4932, *Enterococcus* sp. strain BAB-4933 and *Citrobacter* sp. strain BAB-4934)		65.0%		
	Mixed bacterial culture	Mangrove wetland soils, Thailand	100 mg/L,	pH 8.0, 30 °C, 0.5% (*w*/*v*) NaCl, 72 h	This study
			94.2%		
**C.I. Reactive Blue 19 (RBBR)**	*Bacillus* sp.	Mountain soils, Japan	100 mg/L,	pH 6.0, 30 °C,	[30]
	strain CS-1		100.0%	72 h	
	*Bacillus* sp.	Mountain soils, Japan	100 mg/L,	pH 6.0, 30 °C,	[30]
	strain CS-2		95.0%	72 h	
	*Lysinibacillus sphaericus* strain BR2308	Coastal wetland soils, Thailand	100 mg/L,	pH 6.0, 30 °C,	[19]
			58.1%	72 h	
	*L. sphaericus*	Wetland and estuary soils, Thailand	50 mg/L,	pH 6.0, 28 °C,	[14]
	strain JD1103		50.0%	72 h	
	*Bacillus* sp.	Marsh and grassland samples, South Africa	100 mg/L,	pH 6.0, 25 °C,	[77]
	strain NWODO-3		72.1%	30 min	
	*Enterobacter* sp.	Mangrove wetland soils, Thailand	100 mg/L,	pH 9.0, 30 °C, 3% (*w*/*v*) NaCl, 72 h	This study
	strain RY11903		60.3%		
**C.I. Reactive Green 19**	*Micrococcus glutamicus* strain NCIM-2168	National Chemical Laboratory, India	50 mg/L,	pH 8.0, 37 °C,	[78]
			100.0%	42 h	
	*Acinetobacter* sp.	Activated sludges, Belgium	100 mg/L,	30 °C, 72 h	[79]
	strain ST16.16/164		93.4%		
	Mixed bacterial culture	Mangrove wetland soils, Thailand	100 mg/L,	pH 8.0, 30 °C, 0.5% (*w*/*v*) NaCl, 72 h	This study
			92.1%		
**C.I. Reactive Red 195**	*Georgenia* sp.	India	50 mg/L,	pH 7.0, 40 °C, 5 h	[80]
	strain CC-NMPT-T3		95.9%		
	*Enterococcus faecalis*	Dye contaminated soils, India	50 mg/L,	pH 5.0, 40 °C,	[81]
	strain YZ66		99.5%	1.5 h	
	Mixed bacterial culture	Mangrove wetland soils, Thailand	100 mg/L,	pH 8.0, 30 °C, 0.5% (*w*/*v*) NaCl, 72 h	This study
			84.3%		

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
