# Peer review of "Klebsiella and Enterobacter Isolated from Mangrove Wetland Soils in Thailand and Their Application in Biological Decolorization of Textile Reactive Dyes"

_ijerph, 2020, doi:10.3390/ijerph17207531_

Round 1
Reviewer 1 Report
Dear Author,
please find my comments below.
The revised manuscript aims checking the ability of 4 bacterial strains isolated from Thailand mangrove for decolourization of 9 reactive textile dyes. The manuscript is interesting tackling the important problem of water pollution and the need of it purification. The bacteria were identified and exposed to given doses of dye showing various efficiency. In addition, the efficiency of monoculture versus consortium was studied.
From this point of view I found revised manuscript valuable and interesting and I can state that it falls into the scope of International Journal of Environmental and Public Health. The title is very informative indicating the objectives; abstract provides sufficient number of information about the problem, aim, work done and key results. Keywords are adequate to the topic.
- I would suggest to write “Enterobacter” to indicate the usage of different strains of Enterobacter genus.
Introduction provides enough key information for understanding the problems and existing solutions. Its length is sufficient. First textile dyes are described, then the problems which they cause. Next part describes the possible methods, emphasizing role of bacteria, for dye removal. Author used bacteria originating from mangrove which ecosystem was described well in next section. Last part of introduction in formulation of the aim.
Material & Methods is well organized and divided into sections. First describes the localization of the mangrove. The maps provided are good and sufficient. Next parts describe all procedures. First sampling then isolation of bacteria and their characterization.
- I think that providing the composition of BSGYP medium would be beneficial for this manuscript.
- Line 188: please change „in dark conditions” into „in dark”
The procedures of screening, growth characterization and identification of dye decolouring bacteria are well described and conducted.
- Line 129-130: please change “were ranged” into “ranged”
- Line 132: please give the name, model and manufacturer of spectrophotometer used.
- Line 141: please change “The PCR condition was conducted from” into “The PCR reaction was conducted according to”
Next step of study was enzymatic assays and decolourization efficiency. I found them also well conducted and described. The last part of methodology are statistical procedures.
- It is good manner to mention if normality was checked and what if normality was not met. In addition, the accepted significance (p-value) should be mentioned.
Results and discussion is a main part of the manuscript. Author describes all observations step-by-step in the order similar to methodology section. All data are presented in tables and figures followed by text description. I found them very valuable and clear. It was very clear message when screening immediately revealed only 4 out of 190 were able to decolourized RB19. Next step allowed find optimal growth conditions in terms of pH, temperature and salinity for 4 studied isolates. Genetic analysis was presented in table and on phylogenetic tree. It is also well done. Next sections describe enzymatic activities and dye removal efficiency and the comparison with literature data. These are key data showing clear effects of studied isolates for dye decolourization in comparison for control, E. coli. There is also reliable comparison with the literature done (Table 6). All these data are accurate and sound leading to clear conclusions.
Reviewer 2 Report
This paper reports that four promising bacteria were found from mangrove wetland in Thailand and these bacteria were subjected to decolorization test for artificial textile dyes, and is evaluated to contain some novel findings. However, there are many parts to be modified. Please correct your manuscript according to the comments in the attached file.

Author Response
Please see the attachment.

This manuscript is a resubmission of an earlier submission. The following is a list of the peer review reports and author responses from that submission.